# The structure of the core NuRD repression complex provides insights into its interaction with chromatin

Christopher J Millard[1], Niranjan Varma[1], Almutasem Saleh[1], Kyle Morris[2], Peter J Watson[1], Andrew R Bottrill[3], Louise Fairall[1], Corinne J Smith[2], John WR Schwabe[1]*

[1]Henry Wellcome Laboratories of Structural Biology, Department of Molecular and Cell Biology, University of Leicester, Leicester, United Kingdom; [2]School of Life Sciences, University of Warwick, Coventry, United Kingdom; [3]Protein and Nucleic Acid Chemistry Laboratory, Core Biotechnology Services, University of Leicester, Leicester, United Kingdom

**Abstract** The NuRD complex is a multi-protein transcriptional corepressor that couples histone deacetylase and ATP-dependent chromatin remodelling activities. The complex regulates the higher-order structure of chromatin, and has important roles in the regulation of gene expression, DNA damage repair and cell differentiation. HDACs 1 and 2 are recruited by the MTA1 corepressor to form the catalytic core of the complex. The histone chaperone protein RBBP4, has previously been shown to bind to the carboxy-terminal tail of MTA1. We show that MTA1 recruits a second copy of RBBP4. The crystal structure reveals an extensive interface between MTA1 and RBBP4. An EM structure, supported by SAXS and crosslinking, reveals the architecture of the dimeric HDAC1: MTA1:RBBP4 assembly which forms the core of the NuRD complex. We find evidence that in this complex RBBP4 mediates interaction with histone H3 tails, but not histone H4, suggesting a mechanism for recruitment of the NuRD complex to chromatin.

*For correspondence: john. schwabe@le.ac.uk

**Competing interests:** The authors declare that no competing interests exist.

## Introduction

The nucleosome remodelling and deacetylase (NuRD) complex is one of at least five corepressor complexes that recruits and activates class I histone deacetylase (HDACs) enzymes and directs their activity to chromatin (*Bantscheff et al., 2011*; *Guenther et al., 2000*; *Humphrey et al., 2001*; *Itoh et al., 2015*; *Kelly and Cowley, 2013*; *Laherty et al., 1997*; *Oberoi et al., 2011*; *Vermaak et al., 1999*; *Watson et al., 2012*; *Wen et al., 2000*; *Xue et al., 1998*; *Zhang et al., 1999*). The NuRD complex plays a major role in the regulation of gene expression as well as being involved in chromatin organization, DNA damage repair, and genomic stability (*Denslow and Wade, 2007*; *Lai and Wade, 2011*). The NuRD complex is composed of six proteins, each with several isoforms: HDAC1/2, MTA1/2/3, RBBP4/7, P66α/β, MBD2/3, and CHD3/4 (*Allen et al., 2013*). The complex is special in its ability to exhibit dual enzymatic functionality through its HDAC (protein deacetylation) and ATPase (chromatin remodelling) domains.

Histone Deacetylases (HDACs) are important regulators of transcription that are involved in silencing genes and/or priming them for successive rounds of transcription (*Micelli and Rastelli, 2015*; *Verdin and Ott, 2014*; *Wang et al., 2009*). HDACs act by removing the acetyl groups from modified lysine residues on histone tails and other non-histone proteins. These enzymes have been shown to be overexpressed in various cancer tissues and several inhibitors against these enzymes

**eLife digest** The correct regulation of our genes is essential for life. Genes are actively switched on or off through the action of assemblies of proteins that act together as molecular machines. Some of these machines alter the way that DNA is packaged inside cells. Packaged DNA – called chromatin – consists of DNA wrapped around proteins called histones, which together form structures called nucleosomes. Changing how tightly nucleosomes are packed together can alter whether a gene is active: tighter packing makes it harder to access the genes in that stretch of DNA and therefore inactivates them.

In humans, an assembly of proteins called the NuRD complex makes chromatin more compact by removing acetyl groups from nucleosomes. This complex is important for early development and for the stability and repair of our genes. Three proteins make up its core: HDAC1, which removes the acetyl group from the nucleosome; MTA1, which acts as a scaffold to hold the complex together; and RBBP4, which enables the complex to interact with nucleosomes.

Understanding how protein complexes are assembled tells us a lot about how they work. Millard et al. have therefore used a number of structural techniques to investigate the three-dimensional architecture of the three core proteins in the NuRD complex. The resulting structures have revealed how the HDAC1, MTA1 and RBBP4 proteins interact to influence how the complex is recruited to nucleosomes. The next step will be to assemble all the remaining proteins of the NuRD complex to understand its architecture as a whole.

are currently used in the clinic as anti-cancer therapeutics (*Falkenberg and Johnstone, 2014*; *Marks and Xu, 2009*).

Metastasis-associated protein 1 (MTA1) (and its closely related homologues MTA2 and 3) are up-regulated in various cancer tissues (*Fujita et al., 2003*; *Li et al., 2012*; *Sen et al., 2014*; *Toh et al., 1994*). They serve as scaffold proteins for the assembly of the NuRD complex. The amino-terminal region contains four established domains: an amino-terminal BAH domain, followed by ELM2 and SANT domains, and a GATA-like zinc finger domain towards the centre of the protein (*Manavathi and Kumar, 2007*; *Millard et al., 2014*). The ELM2 domain has been shown by X-ray crystallography to mediate dimerization of the complex and, in conjunction with the SANT domain, recruits HDAC1 & 2 (*Millard et al., 2013*). The structure and function of the MTA-BAH and zinc finger domains remain to be determined. The carboxy-termini of the MTA proteins are much more diverse and structure predictions suggest that much of the protein is intrinsically disordered. The exception is at the very carboxy-terminus of MTA1 and MTA2 where a short helix has been shown to mediate interaction with RBBP4/7 (*Alqarni et al., 2014*). A binding site for CHD4 has also been identified in this region, but has not been structurally characterised (*Nair et al., 2013*).

RBBP4 and RBBP7 are homologous (92% identical) histone binding proteins (also known as RbAp48 and RbAp46) that have been shown to be core components of the NuRD complex. These proteins are unusual in that they are present in multiple complexes including other class I HDAC corepressors such as the Sin3A and PRC2 complexes (*Ciferri et al., 2012*; *Kuzmichev, 2002*; *Zhang et al., 1997*). These proteins have also been reported to have roles in nucleosome assembly: RBBP4, along with sNASP and Asf1 forms a multi chaperone complex whose function is to maintain a supply of newly synthesized histone H3 and H4 proteins under replicative stress (*Groth et al., 2005*). RBBP4 has also been found to be a subunit of the chromatin-assembly factor-1 (CAF-1) complex whose function is to initiate nucleosome assembly by assembling histones H3 and H4 onto newly synthesized DNA (*Parthun et al., 1996*; *Verreault et al., 1996*). In addition, RBBP7 has been found to be an essential subunit of the HAT1 histone-acetyltransferase complex, which acetylates newly synthesized histone proteins (*Verreault et al., 1998*). RBBP4 and RBBP7 have been shown to interact directly with histone H3 and H4 tails through two distinct binding sites and therefore have the potential to act as chromatin recruitment modules (*Murzina et al., 2008*; *Nowak et al., 2011*; *Schmitges et al., 2011*; *Song et al., 2008*; *Zhang et al., 2013*).

RBBP4 and RBBP7 are WD40 domain proteins with a characteristic seven bladed β-propeller structure (*Xu and Min, 2011*). RBBP4/7 interacts with histone H3 through a binding site on the 'top'

face of the protein (*Schmitges et al., 2011*). A particular feature of the RBBP4/7 WD40 domain is an amino-terminal helix of 30 residues that forms an extension to the β-propeller domain. A groove that is formed adjacent to this helix mediates interaction with the amino-terminus of histone H4 (*Murzina et al., 2008*). Interestingly, the extreme carboxy-terminus of MTA1 (residues 671–690) has been shown to interact with RBBP4/7 in this groove, making very similar contacts to those made by histone H4 (*Alqarni et al., 2014*).

In this study, we have identified a second RBBP4 recruitment site within MTA1 that is positioned near the centre of the corepressor. This site shows some sequence homology to the previously characterised carboxy-terminal recruitment site, but is considerably more extensive. We show that both sites within the MTA1 monomer are capable of independently recruiting RBBP4 proteins such that the HDAC1:MTA1 dimer recruits four molecules of RBBP4 to the NuRD complex. The crystal structure of this central recruitment site reveals an extensive interface that requires some structural rearrangement within RBBP4. Small angle X-ray scattering (SAXS) and single-particle negative-stain electron microscopy (EM) reveal the architecture of the core NuRD complex. RBBP4 proteins are tethered to the HDAC1 dimer by MTA1 forming an elongated zig-zag conformation. We show that when in complex with MTA1, RBBP4 is able to interact with histone H3, but not histone H4 suggesting a mechanism for recruitment of the NuRD complex to chromatin.

## Results

### The MTA1 dimer recruits four RBBP4 subunits to the NuRD complex

During purifications of various HDAC1:MTA1 complexes expressed in mammalian cells, we observed an endogenous protein of c. 50kDa co-purifying with these complexes. Peptide mass spectrometry identified this protein as being the WD40 protein RBBP4 and/or RBBP7. The interaction between RBBP4/7 and the carboxy-terminus of MTA1 (residues 670–691) is well characterised and an atomic resolution structure of a 22 residue MTA1 peptide bound to RBBP4 has been previously solved by X-ray crystallography (*Alqarni et al., 2014*). However, we were intrigued to find that RBBP4/7 was recruited to constructs that lacked this well-characterised interaction domain. This appears to fit with previous findings that indicated that RBBP4/7 may interact with a region containing the MTA1 zinc finger domain and adjacent sequences (*Roche et al., 2008*).

These findings suggested that two regions of MTA1 are able to mediate interaction with RBBP4. What remained unclear was whether these two regions collaborate to bind to a single RBBP4 protein or whether there are two separate binding sites that result in two RBBP4 proteins binding to MTA1.

To define the biologically relevant regions of MTA1 that recruit RBBP4/7, various fragments of MTA1, and full-length HDAC1, were co-transfected into mammalian cells (*Figures 1a and 1b*). These complexes were purified utilising a FLAG tag on MTA1 and we examined the apparent stoichiometry for the recruitment of endogenous RBBP4/7 to these complexes using gel densitometry (*Figure 1c*). Importantly, the MTA1-B construct spanning residues 162–546 appears to pull down a stoichiometric 1:1 ratio of HDAC1:RBBP4/7 suggesting that this region is sufficient to form a bona fide interface with RBBP4/7. Interesting a construct spanning residues 162–715 (MTA1-C) pulls down a supra-stoichiometric ratio of 1:1.5 of HDAC1:RBBP4/7. This implies that there may be two RBBP4/7 proteins associated with MTA1 (*Figure 1c*).

To confirm the stoichiometry of the complex we co-expressed RBBP4 with different length MTA1 fragments extending from the zinc finger domain. The resulting complexes were purified by gel filtration and analysed with Multi-Angle Light Scattering (MALS) to determine the overall molecular weight. The longer fragment of MTA1-D (residues 390–715) in complex with RBBP4 was shown to have a molecular weight of 130 kDa (*Figures 1d and 1f*; see also *Figure 1—figure supplement 1*). This closely fits with the predicted molecular weight (131 kDa) of a 1:2 stoichiometry of one MTA1 fragment bound to two RBBP4 proteins. The shorter MTA1-R1 fragment (residues 464–546) in complex with RBBP4 was determined to have a molecular weight of 54 kDa which is close to the predicted molecular weight of 58 kDa for the 1:1 complex (*Figures 1e and 1g*; see also *Figure 1—figure supplement 1*).

It is clear that these experiments support a model in which two RBBP4 proteins are recruited by MTA1 to the NuRD complex. Since there are two recruitment sites for RBBP4 we have called these R1 (residues 464–546) and R2 (residues 670–691).

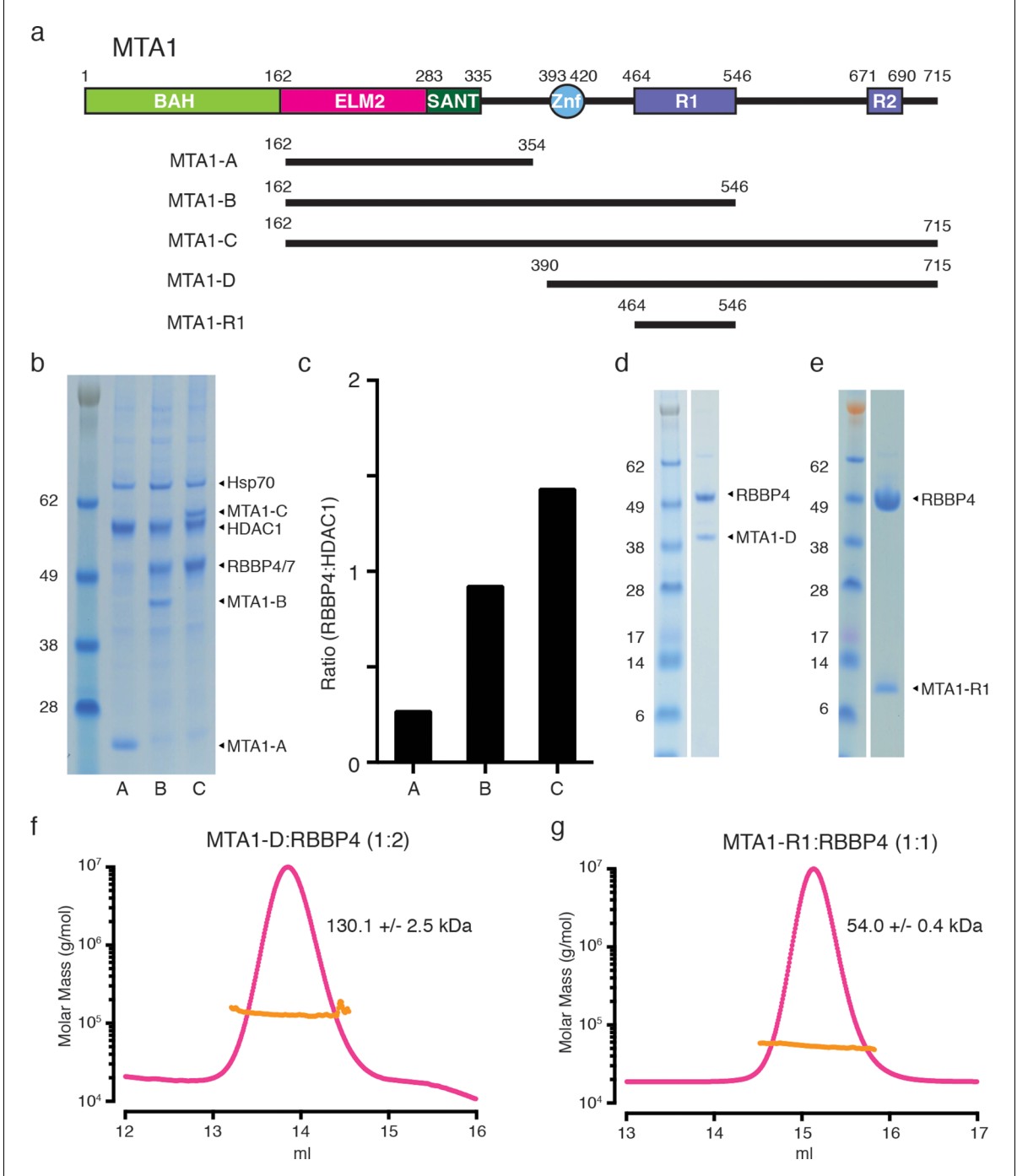

**Figure 1.** MTA1 co-purifies with endogenous RBBP4/7 in a supra-stoichiometric ratio. (a) Schematic representation of the domain structure of MTA1, with the R1 and R2 RBBP4 recruitment domains shown in purple. A summary of fragments used in the interaction studies is shown below. (b) MTA1-B and MTA1-C co-purify with endogenous RBBP4/7, as identified by mass spectrometry, in a stoichiometric and a supra-stoichiometric ratio respectively. (c) The ratio of endogenous RBBP4/7 to co-transfected HDAC1 is quantified from the SDS-PAGE gel by densitometry. (d) and (e) Co-expression of the MTA1-D:RBBP4 and MTA1-R1:RBBP4 complexes. (f) and (g) Co-expressed MTA1-D:RBBP4 and MTA1-R1:RBBP4 are shown to form complexes of 1:2 and 1:1 stoichiometry, respectively, as determined by size exclusion chromatography coupled to multi-angle light scattering (MALS). See *Figure 1—figure supplement 1* for information about expression, purification and crystallisation.

The following figure supplement is available for figure 1:

**Figure supplement 1.** Purification of MTA1:RBBP4 complexes.

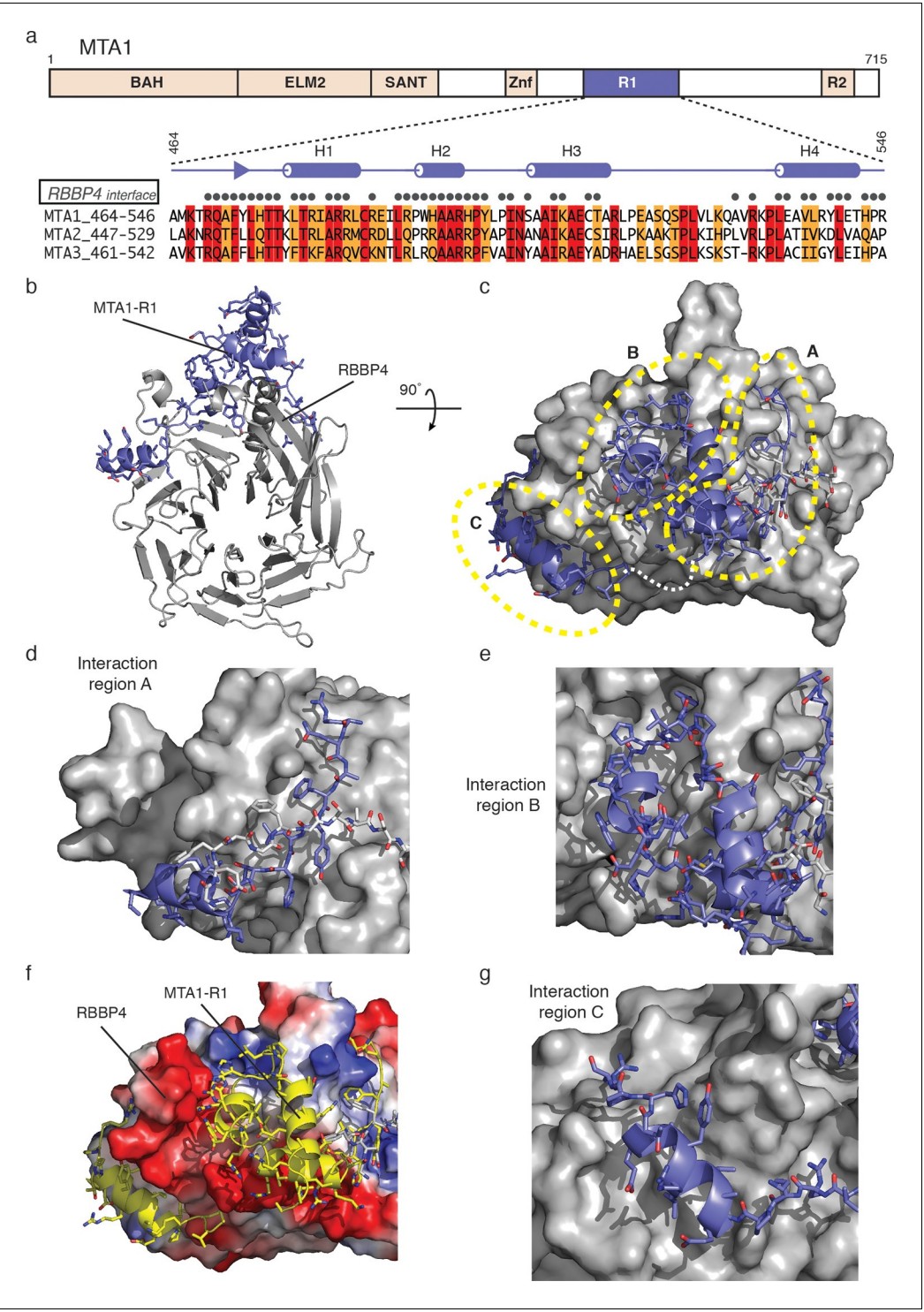

**Figure 2.** The crystal structure of MTA1-R1 domain bound to RBBP4. (a) Schematic representation of the domain structure of MTA1. The secondary structure of MTA1-R1 (464–546), corresponding to the R1 domain, is shown. Those residues of MTA1 that form an interface with RBBP4 in the crystal structure are indicated. Below is the sequence alignment of the R1 domains from MTA1, 2 and 3. Residues coloured red are identical and residues coloured orange are conserved. (b) Cartoon representation of the MTA1-R1:RBBP4 complex with MTA1 in purple and RBBP4 in grey. (c) The structure is rotated by 90° and the surface of RBBP4 is shown. The dotted white line indicates the part of MTA1 that is disordered. The 5G loop in RBBP4 is shown as sticks with grey coloured carbon atoms. MTA1-R1 is rationalised into three interaction regions A, B and C and these are shown in more detail in (d),

*Figure 2 continued on next page*

*Figure 2 continued*

(e) and (g). MTA1 is shown as a cartoon and RBBP4 as a grey surface. (f) Electrostatic surface view of RBBP4 with MTA1 shown in yellow. See *Video 1* for a 3D view of the MTA1-R1:RBBP4 complex and *Table 1* for the crystallographic data collection and refinement statistics.

## An extensive interface is formed between RBBP4 and the R1 domain of MTA1

To understand the nature of the interface between the R1 domain from MTA1 and RBBP4 we co-expressed and purified a complex between MTA1-R1 (residues 464–546) and full-length RBBP4. The complex was affinity purified on Flag resin using an amino-terminal tag on MTA1 before being cleaved from the resin and gel filtrated. The complex was concentrated to 10 mg/ml and small crystals were obtained. X-ray diffraction data were collected to 3.2 Å and the structure solved by molecular replacement using the structure of RBBP4 (pdb code: 4PC0). The density for MTA1 was remarkably clear and the structure of the complex was built through multiple rounds of refinement (*Figure 1—figure supplement 1*).

The seven bladed beta-propeller fold of RBBP4 is largely unchanged by the binding of MTA1-R1. The first 10 residues of RBBP4 are unstructured, and similar to previous structures of RBBP4/7, the 14 carboxy-terminal residues are not observed. A notable difference in this structure is the stabilisation of a loop that supports binding of MTA1 (see below).

The interface between MTA1-R1 and RBBP4 is extensive with a buried surface area of 2,837 Å$^2$ (*Figures 2a and 2b*, *Table 1* and *Video 1*). MTA1 interacts on the side of the WD40 beta-propeller domain making interactions in three adjacent grooves. Two of these grooves lie on either side of the long amino-terminal helix that forms an extension to the WD40 domain. The contacts made by MTA1 on the surface of RBBP4 can be rationalised into three regions hereafter termed interaction regions A, B and C (*Figure 2c*). The residues involved in all three interaction regions are conserved in both MTA2 and MTA3 (*Figure 2a*).

The interaction region A of MTA1 takes the form of an extended strand followed by a short helix (H1) (*Figure 2d*). The extended strand positions a number of non-polar sidechains into a groove formed between the amino-terminal helix and adjacent strand of RBBP4 that forms part of the 'linking' blade of the WD40 domain. This region of MTA1 is held in place by a loop within RBBP4 that is an insert in blade one (residues 104–113) of the WD40 domain. This loop contains five glycine residues and is referred to as the 5G-loop. The 5G-loop adopts an ordered structure on MTA1 binding such that several non-polar residues form an additional interface with MTA1.

Helix H1 of interaction region A is immediately carboxy-terminal to the extended strand and is oriented perpendicularly to it. The helix lies across the base of the long amino-terminal helix of RBBP4 (*Figure 2d*). The interaction involves four arginine residues forming a zipper-like series of salt bridges with three interdigitating acidic residues on the surface of RBBP4.

The central interaction region B of MTA1 consists of a short helix H2, a structured PYxPI loop and a longer helix H3 (*Figure 2e*). The helix H2 and PYxPI loop lie in a deep groove between the amino-terminal helix of RBBP4 and the so-called PP-loop (which contains two adjacent prolines). The groove is largely non-polar in character, but the periphery is strongly acidic (*Figure 2f*). This complements the interacting amino acids in helix H2 and the PYxPI loop of MTA1. Helix H3 interacts with two phenylalanines on RBBP4; F30 on the surface of the amino-terminal helix and F105 with the 5G-loop.

The final interaction region C is composed primarily of helix H4 which binds in a third

**Video 1.** The crystal structure of MTA1-R1 bound to RBBP4 MTA1 (residues 464-546) interacts in three adjacent grooves on the side of RBBP4. RBBP4 is shown as surface (grey) and MTA1 as cartoon (purple). This video relates to *Figure 2*.

**Table 1.** Crystallographic data collection and refinement statistics.

| Data Collection | MTA1-R1:RBBP4 complex |
|---|---|
| Beamline | Diamond I24 |
| Space group | P1 $2_1$ 1 |
| Wavelength, Å | 0.96861 |
| Cell dimensions | |
| a, b, c, Å | 81.29, 150.07, 95.59 |
| α, β, γ, ° | 90, 94.54, 90 |
| Resolution range, Å* | 95.29–3.2 (3.34–3.2) |
| $R_{merge}$* | 0.165 (0.805) |
| $R_{meas}$* | 0.176 (0.872) |
| $CC_{1/2}$* | 0.975 (0.552) |
| Mean I/σI* | 5.3 (2.0) |
| Completeness, %* | 97.7 (98.0) |
| Multiplicity* | 4.2 (4.2) |
| Refinement | |
| Resolution range, Å | 95.29–3.2 |
| No. of reflections | 35,054 |
| $R_{work}/R_{free}$ | 0.249/0.291 |
| Number of atoms | |
| Protein | 14345 |
| Water | 0 |
| B-factors, Å$^2$ | |
| Protein | 90.7 |
| Rmsd from ideal values | |
| Bond lengths, Å | 0.007 |
| Bond angles, ° | 1.053 |
| Ramachandran plot | |
| Favoured, % | 94.7 |
| Allowed, % | 4.6 |
| Outliers, % | 0.7 |
| Missing residues | MTA1, 464–467, 519–528 |
| | RBBP4, 1–10, 90–103, 176–179, 412–425 |

*The highest resolution shell is shown in parentheses.

groove on the surface of RBBP4 on the far side of the PP-loop (**Figure 2g**). The interaction is primarily non-polar in character.

## Architecture of the core NuRD complex

Bioinformatic analysis of MTA1 indicates that there are only short regions of predicted disorder between the ELM2-SANT domain and the R1 domain. This suggested that the MTA1:HDAC1:RBBP4 proteins might form a structural core to the NuRD complex with a fixed arrangement of subunits. We have used a combination of small angle X-ray scattering (SAXS), single-particle negative-stain electron microscopy (EM) and chemical crosslinking to determine the architecture of this core complex. For the SAXS analysis, we compared the scattering profile and the calculated envelope of the

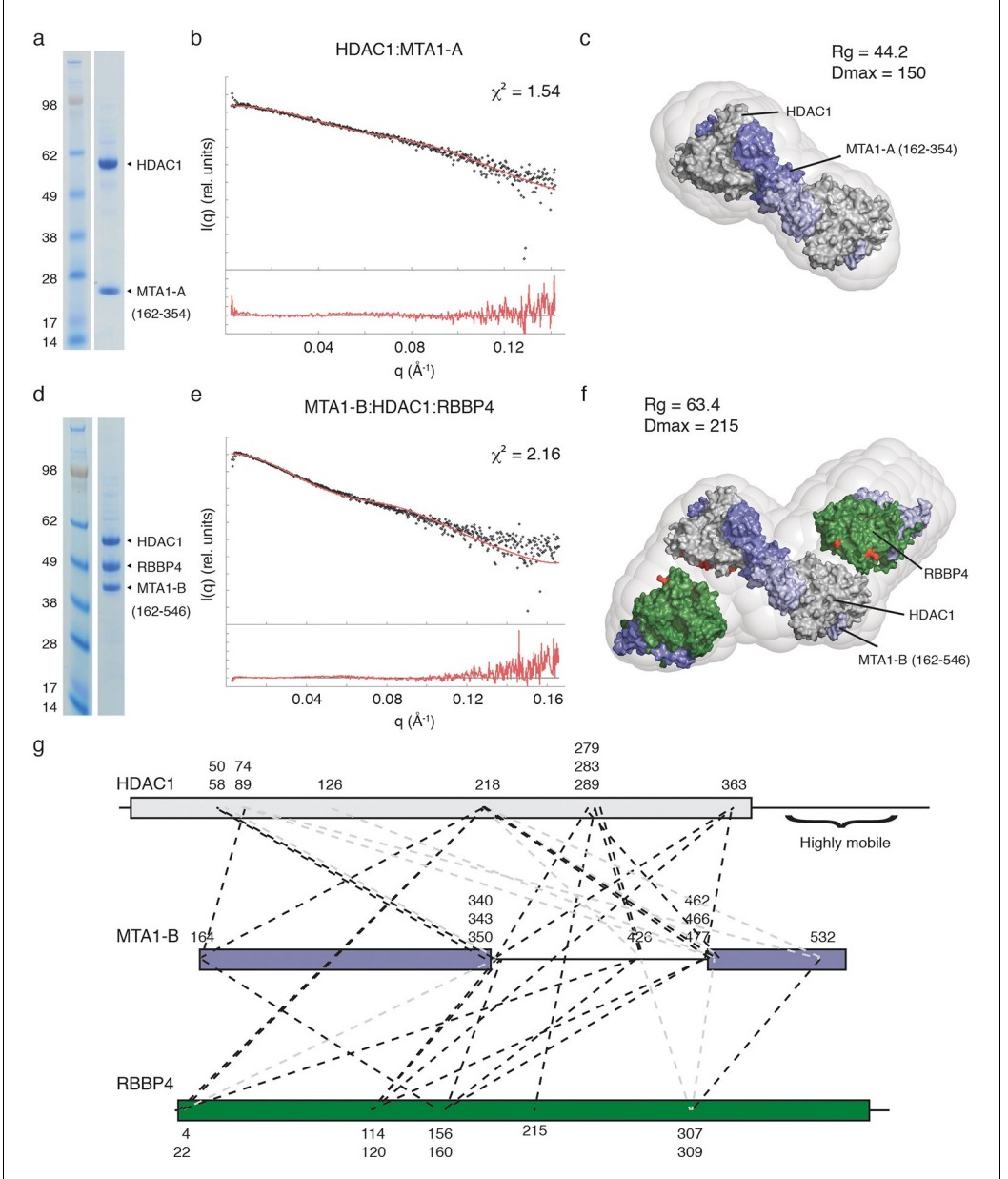

**Figure 3.** The core NuRD complex has an elongated structure. (**a**) Co-expression of the HDAC1:MTA1-A (162–354) complex. (**b**) SAXS data of HDAC1:MTA1-A with the theoretical scattering curve from the HDAC1:MTA1 dimer (pdb code: 4BKX) superimposed in red and the fit residuals are shown. (**c**) The HDAC1:MTA1 crystal structure is fitted into the *ab initio* molecular envelope derived from the SAXS curve. MTA1 is coloured purple and HDAC1 is shown in grey. (**d**) Co-expression of the MTA1-B (162–546):HDAC1:RBBP4 complex. (**e**) SAXS data of MTA1-B: HDAC1:RBBP4 with the theoretical scattering curve from the MTA1:HDAC1:RBBP4 model superimposed in red and the fit residuals are shown. (**f**) The model of MTA1:HDAC1:RBBP4 is fitted into the *ab initio* molecular envelope derived from the SAXS curve. MTA1 is shown in purple, HDAC1 in grey and RBBP4 in green. The MTA1-R1:RBBP4 crystal structure is orientated based on crosslinks identified on surface of HDAC1 and RBBP4 and these are coloured red. (**g**) Isotopic crosslinking of MTA1-B:HDAC1:RBBP4. Black dotted lines indicate crosslinks that could be mapped onto the model of MTA1:HDAC1:RBBP4 and those that could not be mapped are shown in grey. The HDAC1 carboxy-terminus is predicted highly disordered and the large number of crosslinks seen to this region suggest it is highly mobile. These crosslinks are not shown for simplicity. See *Figure 3—figure supplement 1* and *Figure 3—figure supplement 2* for information about purification and SAXS measurements, and *Figure 3—source data 1* for crosslinks between MTA1, HDAC1 and RBBP4.

*Figure 3 continued on next page*

*Figure 3 continued*

The following source data and figure supplements are available for figure 3:

**Source data 1.** Crosslinks within the MTA1-B (162–546):HDAC1:RBBP4 complex.
**Figure supplement 1.** Purification and SAXS analysis of the HDAC1:MTA1-A (162–354) complex.
**Figure supplement 2.** Purification and SAXS analysis of the MTA1-B (162–546):HDAC1:RBBP4 complex.

HDAC1:MTA1-A (residues 162–354) complex with that of the MTA1-B (residues 162–546):HDAC1:RBBP4 complex.

Multi-Angle-Light-Scattering (MALS) confirmed that, as expected from the crystal structure, the HDAC1:MTA1 complex is dimeric, containing two copies of each protein (*Figure 3—figure supplement 1*). The SAXS data for the complex are consistent with minimal aggregation (linear I(q) at low q). A theoretical scattering profile was calculated from the crystal structure of HDAC1:MTA1 (*Millard et al., 2013*) and this was in good agreement with the experimental data with a $\chi^2$ function of 1.54. Furthermore, the crystal structure of the HDAC1:MTA1 dimer was a convincing fit to the averaged *ab initio* molecular envelope generated from the SAXS curve (*Figures 3a, 3b and 3c*).

MALS analysis of the MTA1:HDAC1:RBBP4 complex showed that this ensemble is also dimeric, with the experimental mass of 300 kDa closely matching the theoretical mass (294 kDa) (*Figure 3—figure supplement 2*). The *ab initio* molecular envelope generated from the SAXS curve in the lowest symmetry mode (P1) suggests that the core NuRD complex has an elongated architecture (*Figures 3d,e and f*). Although no symmetry was imposed, the envelope has an approximate two-fold symmetry consistent with a dimeric complex. We fitted the HDAC1:MTA1 dimer in the centre of the envelope such that the two-fold axis of symmetry matched the approximate symmetry of the envelope (*Figure 3f*). This left two 'lobes' to the envelope that were of sufficient volume to contain the structure of the MTA1-R1:RBBP4 complex and additional volume for the zinc-finger domain that forms a link between the two X-ray structures.

Our crosslinking data indicated that one face of HDAC1 could readily form crosslinks with one face of RBBP4, suggesting a particular relative orientation of the two proteins in the core complex. We orientated the MTA1-R1:RBBP4 structure within the two 'lobes' so as to satisfy the largest number of crosslinks (*Figure 3g* and *Figure 3—source data 1*). The crosslinks that could not be satisfied are likely to derive from higher order oligomers of the core complex. A theoretical scattering profile was calculated from the docked model of the MTA1:HDAC1:RBBP4 complex. This showed reasonable agreement with the experimental data with a $\chi^2$ function of 2.16.

To further investigate the architecture of the MTA1:HDAC1:RBBP4 ensemble we used negative stain EM to determine a low resolution structure of the core NuRD complex, crosslinked with glutaraldehyde to maintain integrity. A 60 Å Gaussian low-pass-filtered, initial model was generated based on the model derived from the SAXS, crosslinking and crystal structure data. This initial model was refined against 17,841 semi-automatically picked particles (*Figure 4a*). The resulting EM envelope showed that RBBP4 and HDAC1 form a closer association than predicted in the SAXS model. The HDAC1:MTA1 and MTA1-R1:RBBP4 structures fitted well within this envelope. In their new position the RBBP4 domains are tipped by 30° such that they are approximately parallel to each other and propagate the symmetry of the HDAC1:MTA1 dimer. This position fits well with the cross-linking data. A 30 Å filtered model was then generated, based on the new position of RBBP4, and refined further to produce a 19 Å EM structure using the FSC criterion with a cutoff of 0.143 (*Figure 4b,c*).

The resulting EM envelope, along with the fitted crystal structures, clearly shows the HDAC1:MTA1 central dimer and a detailed outline of the RBBP4 lobes (*Figure 4c* and *Video 2*). The model suggests a number of contacts between RBBP4 and HDAC1 that determine the relative orientation. The active sites of both HDACs are on the convex side of the complex and are exposed to enable substrate binding. Extra density surrounding the dimerization domain of MTA1 may indicate the position of the MTA1 zinc finger domains and/or the tails of HDAC1 and RBBP4. The reference-free 2D class averages fit well with the model of the MTA1:HDAC1:RBBP4 complex and with the re-projections of the EM class averages (*Figure 4d* and *Figure 4—figure supplement 1*).

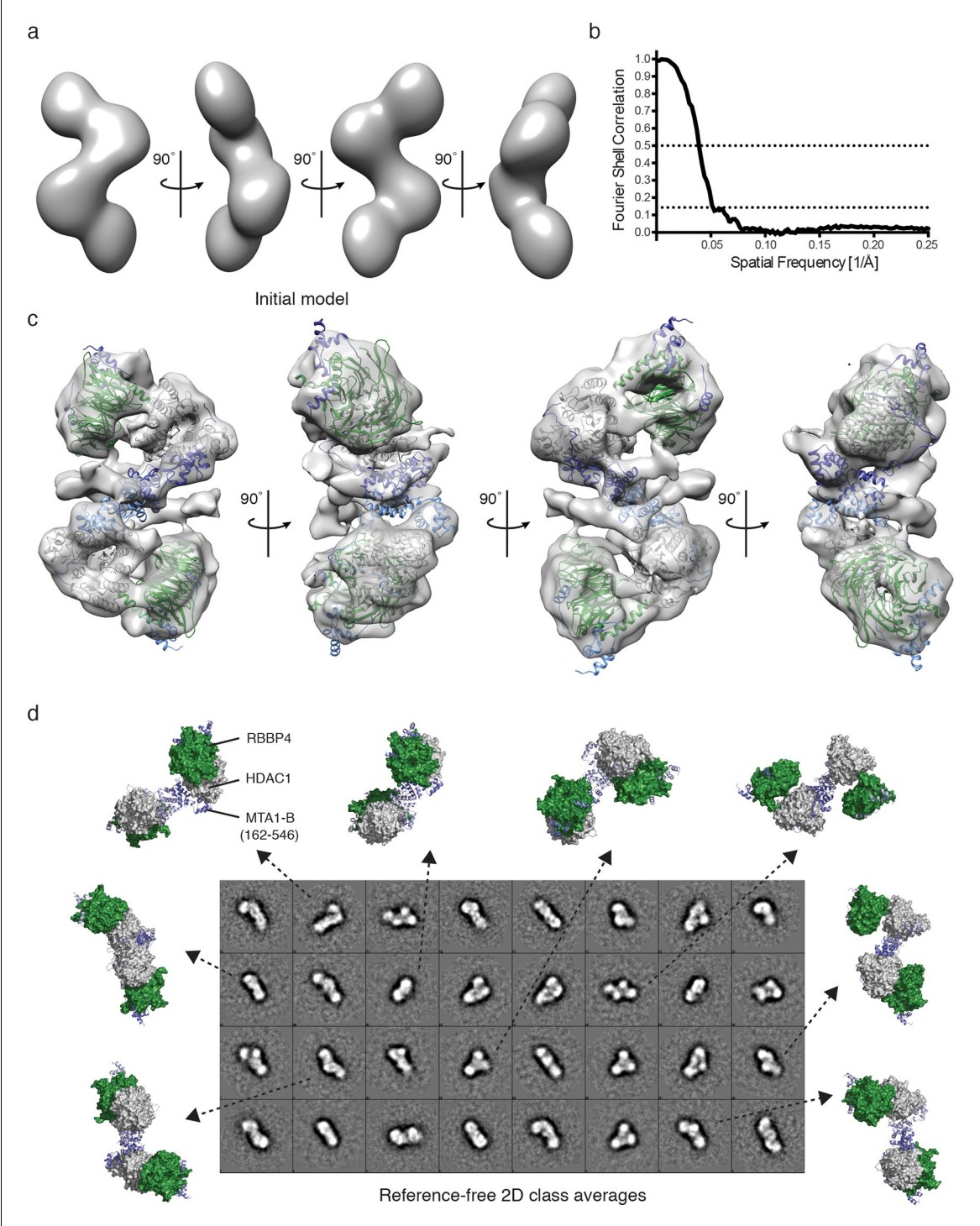

**Figure 4.** Visualisation of the core NuRD complex by negative-stain electron microscopy. (**a**) 60 Å filtered initial model of the MTA1-B (162–546):HDAC1: RBBP4 complex generated from the SAXS and crosslinking data. (**b**) The resolution of the final EM model was estimated to be 19 Å using the Fourier shell correlation criterion with a cutoff of 0.143 (26 Å using a cutoff of 0.5). (**c**) Final EM model and fitting after refinement against 17,841 particles. (**d**) Comparison of different views of the MTA1-B:HDAC1:RBBP4 complex, generated after EM model fitting, with the reference-free 2D class averages. MTA1 is shown in blue, HDAC1 in grey and RBBP4 in green. See *Video 2* for a 3D view of the MTA1-B (162–546):HDAC1:RBBP4 complex and *Figure 4—figure supplement 1* for class averaged particles and re-projections of the EM model.

The following figure supplement is available for figure 4:

*Figure 4 continued on next page*

*Figure 4 continued*

**Figure supplement 1.** Further analysis of the EM model.

## Comparison with other RBBP4 complexes

Since we have established that two copies of the RBBP4 protein are recruited to MTA1, we compared the amino-terminal recruitment domain, R1, with the previously identified carboxy-terminal domain, R2. Sequence alignment show that a stretch of c.35 residues is highly similar between the two recruitment regions (*Figure 5a*). These include the amino terminal strand, helices H1 & H2 and the PYxPI loop. A previous structure of the R2 domain in complex with RBBP4 included helix H2 and the PYxPI loop (22 residues) (*Alqarni et al., 2014*). Comparison of this structure with that of the equivalent region of the R1 domain (region B in the description above) shows a highly similar mode of binding to RBBP4 (*Figure 5b*). Indeed, the path of the mainchain from P492 to I503 is essentially identical. The sidechains A495, A496 and R497 in helix H2 and P499, Y500, P502 and I503 in the PYxPI loop mediate key interactions with RBBP4 and are identical in the R2 domain.

The sequence alignment also indicates that the 15 residues preceding helix 2 in the R2 domain have significant sequence similarity to the R1 domain suggesting that the interface between RBBP4 and the R2 domain is likely to be rather more extensive than was observed in the previously reported crystal structure (*Figure 5a*). We would expect these residues to adopt a structure similar to that seen in interaction region A of the complex between RBBP4 and the R1 domain. An important part of this interface is the 5G-loop that wraps over the top of the amino-terminal strand in the R1 domain. Interestingly this loop adopts a very different conformation in the structure of the MTA1-R2:RBBP4 complex. This alternative orientation may be the consequence of crystal packing interactions.

The surface of RBBP4 that mediates interactions with helix H2 of the R1 and R2 domains of MTA1 has also been shown to mediate interactions with the amino-terminal region of histone H4. Sequence alignment of the MTA1-R1 and -R2 domains with histone H4 shows that there is considerable sequence similarity between these proteins. The structure of histone H4 (residues 27–41) in complex with RBBP4 shows a similar mode of binding to the structure of the bound MTA1-R1 and -R2 domains (*Figure 5c*) (*Murzina et al., 2008*). All three proteins bind in the same groove and make similar sidechain contacts with an Alanine-Arginine dipeptide making identical contacts in all three structures (*Figures 5b and 5c*). Again, sequence comparison suggests that the interface between histone H4 and RBBP4 may be more extensive than seen in the previously reported structure, since the sequence similarity extends at the amino-terminus, into the region corresponding to helix H1 of MTA1-R1.

Interestingly, a fragment of the Polycomb protein Su(z)12 has also been crystallised bound to the central groove of RBBP4. There is little structural or sequence similarity to either the histone H4, MTA1-R1 or MTA1-R2 complexes.

## RBBP4 in the NuRD complex binds histone H3 but not histone H4

The previous observation that RBBP4 is able to bind to both histones H3 and H4 suggests that RBBP4 may serve as a common chromatin recruitment module in several chromatin modifying complexes. However, given that the extensive interface between RBBP4 and the R1 domain of MTA1 (and by inference the R2 domain) occludes the binding site for histone H4, we would predict that RBBP4 is no longer able to bind histone H4 in the NuRD complex.

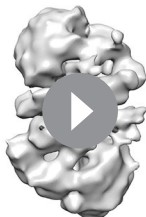

**Video 2.** EM structure of the MTA1-B:HDAC1:RBBP4 complex. Low resolution EM structure (19 Å) of the core NuRD complex is shown contoured at 1. Structures of the dimeric HDAC1:MTA1 (pdb code: 4BKX) and MTA1:RBBP4 (pdb code: 5FXY) are fitted. HDAC1 is shown in grey, RBBP4 in green and MTA1 in blue. This video relates to *Figure 4*.

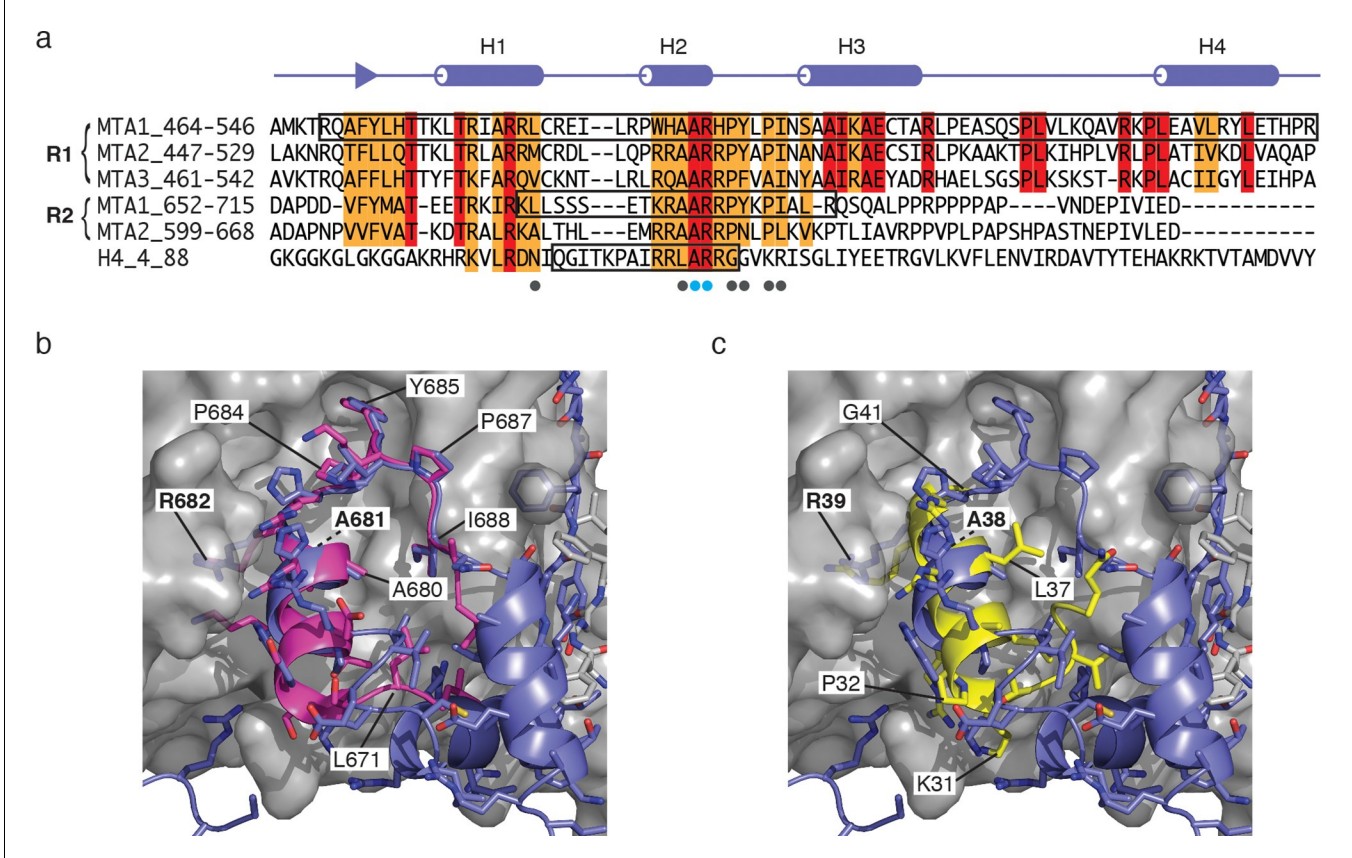

**Figure 5.** Comparison of the MTA1-R1:RBBP4 structure with other RBBP4 complexes. (**a**) The domain structure of MTA1 highlighting the R1 and R2 domains. Below is a sequence alignment of the R1 domains of MTA1/2/3, R2 domains of MTA1/2 and histone H4. Boxes highlight the residues that have been crystallised in complex with RBBP4. Residues coloured red are identical and residues coloured orange are conserved. (**b**) Overlay of crystal structures of MTA1-R1 domain with a peptide corresponding to the MTA1-R2 domain (671–690) in complex with RBBP4. (**c**) Overlay of crystal structures of the MTA1-R1 domain with a histone H4 peptide (27–41) in complex with RBBP4.

However, the histone H3 interaction on RBBP4 is exposed in both the MTA1:RBBP4 binary complex and in the dimeric MTA1:HDAC1:RBBP4 complex. This implies that RBBP4 is likely to play a role in recruiting the NuRD complex to histone H3 tails, but not to histone H4.

To test this, we compared RBBP4 and the MTA1-R1 (residues 464–546):RBBP4 complex in their ability to bind to a peptide array of histone tails, bearing various post-translation modifications. As predicted, we observed that both RBBP4 and the MTA1-R1:RBBP4 complex showed an identical pattern of binding to histone H3 peptides. However, there were significant differences between the binding profiles to histone H4 modified peptides (*Figure 6a*). The free RBBP4 bound multiple histone H4 tails. In contrast, the MTA1-R1:RBBP4 complex showed only binding to two histone H4 peptides and this was not reproduced in the duplicate arrays (*Figure 6—figure supplement 1* and *Figure 6—source data 1*). Taken together this suggests that when bound to MTA1, RBBP4 binds to histone H3 and not histone H4.

## Discussion

Here we have investigated the architecture of the core NuRD complex and derived the first model of the arrangement of three major components MTA1/2, HDAC1/2 and RBBP4/7 (*Figure 6b*). This core complex forms an elongated zig-zag structure as visualised by SAXS and EM, centred around the HDAC1:MTA1 dimerization domain, with RBBP4 'lobes' symmetrically bound to the dimer. We have established that a central section of MTA1 directly recruits RBBP4/7 and brings this subunit into close proximity to HDAC1/2. It is possible that the relative arrangement of HDAC1, MTA1 and

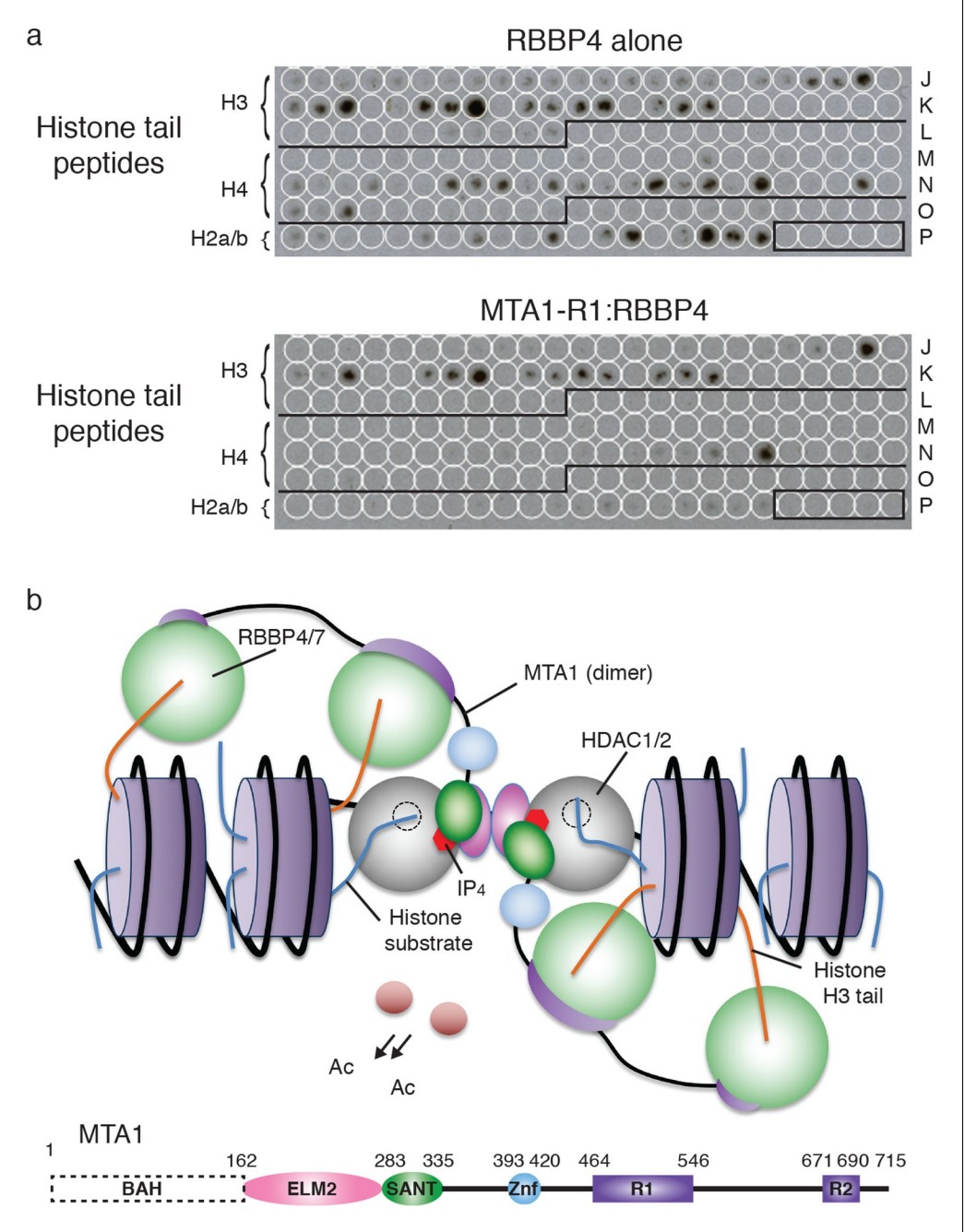

**Figure 6.** Recruitment of the core NuRD complex to chromatin could be mediated by RBBP4 binding to histone H3. (a) RBBP4 binds to both histone H3 and histone H4 tail peptides on the MODified histone peptide array whereas the complex of MTA1-R1:RBBP4 is unable to bind to histone H4 tail peptides. MODified histone peptides J1 to P24 are shown. The five negative control peptides (non-histone sequences) are boxed in the right-hand corner. (b) Schematic model for the recruitment of the core NuRD complex to chromatin. Two HDAC1/2 molcules (grey) and four RBBP4/7 molcules (green) are tethered by the ELM2/SANT and R1/R2 domains of MTA1 respectively. MTA1 is coloured according to the schematic representation below. The RBBP4/7 proteins recruit the complex to chromatin through histone H3 tails (orange) on the same or adjacent nucleosomes (purple). The other histone tails would be available for deacetylation by HDAC1/2. The BAH domain of MTA1 is omitted from the model for clarity. See *Figure 6—figure supplement 1* for the MODified and duplicate histone peptide arrays and *Figure 6—source data 1* for a key to the array.

The following source data and figure supplement are available for figure 6:

*Figure 6 continued on next page*

*Figure 6 continued*

**Source data 1.** Key for the MODified histone peptide array – Rows J1 to P24.
**Figure supplement 1.** MODified histone peptide array.

RBBP4 may be somewhat different in the holo-NuRD complex. However, the relatively close interaction between HDAC1 and RBBP4, seen in the EM structure, suggests to us that this interface is likely to be present in the full complex.

We have shown that each MTA1 protein recruits two RBBP4/7 molcules to the NuRD complex through the R1 and R2 interaction domains. Since MTA1 is itself a dimer in the complex (*Millard et al., 2013*) this means that the NuRD complex contains four copies of the RBBP4/7 proteins, which we have confirmed experimentally. Recognisable R1 and R2 interaction domains are present in MTA1 and 2 proteins from fly to man suggesting that the stoichiometry of the NuRD complex is evolutionarily conserved. Our findings fit partially with a previous study of the NuRD complex using label-free mass spectrometry. In this study they also report a 1:2 MTA1:RBBP4/7 stoichiometry, but in contrast to our previous crystal structure of the MTA1:HDAC1 complex, they find a 1:3 HDAC:MTA stoichiometry (*Smits et al., 2012*).

Alqarni et al. used isothermal titration calorimetry (ITC) to compare the affinity of interaction of the helical region of the R2 domain from MTA1 (656–686) with that of histone H4, which they observed bound in the same groove of RBBP4. The similarity in affinity led them to suggest that there may be competition between MTA1 and histone H4 for binding to RBBP4 (*Alqarni et al., 2014*). Our findings suggest that there is likely to be a more extensive interface between MTA1-R2 and RBBP4 with an additional helix and beta-strand contributing to the interaction. This is in fact supported by cross-linking data (*Kloet et al., 2015*). Given this extended interface we suggest that histone H4 is unlikely to be able to compete with either MTA1-R1 or R2 for binding to RBBP4. Indeed, the MTA1-R1:RBBP4 complex appeared highly stable since the resin-bound complex could be washed with 2 M NaCl and 5% triton-X-100 without significant dissociation (data not shown). Taken together our data suggest that once bound to MTA1, the two RBBP4 molcules would remain permanently assembled as a complex.

Since it appears that the histone H4 binding site on RBBP4 is blocked when associated with MTA1 (either R1 or R2) this raises the question as to the role of RBBP4 in the NuRD complex. Importantly, the crystal structure of RBBP4 bound to a histone H3 tail peptide shows that the interaction surface lies on the 'top' of the WD40 domain (*Schmitges et al., 2011*). This site is not blocked by binding to MTA1-R1. Thus RBBP4 would theoretically remain able to bind to histone H3 tails whilst assembled into the NuRD complex. This fits with our experimental data which confirm that the core NuRD complex can bind histone H3 tails but not H4.

Such a model of recruitment to histone H3 is consistent with the architecture of the MTA1:HDAC1:RBBP4 complex since the orientation of RBBP4 relative to the HDAC1:MTA1 dimer means that the histone H3 binding surface of RBBP4 is exposed, albeit close to the interface between HDAC1 and RBBP4, and therefore able to bind the tail of histone H3. Furthermore, the histone recruitment site is sufficiently close to the catalytic site of HDAC1 to allow other histone tails to access the active site.

Interestingly, both RBBP4 and the RBBP4:MTA1 complex show enhanced binding to modified tails of histone H3 lysine 27 (H3K27) (*Figure 6a*). In particular, H3K27 acetylation and H3K27 methylation appear to promote binding. This would correlate with previous studies that have observed association of the NuRD complex with a specific subset of gene promoters enriched with H3K27me3 (*Reynolds et al., 2012*). In addition, MBD3, a component of the NuRD complex has been reported to localise at H3K27ac and H3K27me3 enriched sites (*Shimbo et al., 2013*). RBBP4 may therefore have an important role in directing the NuRD complex to chromatin bearing these modifications.

A number of studies have suggested that RBBP4 and RBBP7 have different functions. Indeed, it has been suggested that they are associated with different complexes. Our structural data suggest that the residues important for mediating interaction with MTA1 are 100% identical and therefore we would expect both RBBP4 and RBBP7 to be associated with the NuRD complex depending upon relative abundance.

Interestingly, in MTA3 the recruitment domain R2 is not present in any species. This suggests that MTA3 only recruits a single RBBP4/7 molcule. Furthermore, a splice variant of MTA1 (MTA1S) does not contain either R1 or R2 domains (*Yaguchi et al., 2005*). It is intriguing to speculate that these variant NuRD complexes will exhibit different specificities due to different modes of chromatin recruitment. Presumably the presence of four RBBP4 molcules in the NuRD complex increases the avidity for chromatin.

Overall this study provides significant novel insights into the assembly and chromatin recruitment of the core NuRD complex. It establishes the architectural relationship between the histone deacetylase and the RBBP4 chromatin-binding module. Further studies will be required to explore the relative organisation between the HDAC and the CHD3/4 chromatin remodeller.

## Materials and methods

### Mammalian protein expression

MTA1-A, -B, -C, -D and -E (residues 162–335, 162–546, 162–715, 390–715 and 464–546) were cloned into pcDNA3 vectors containing an amino terminal $His_{10}$-$Flag_3$ purification tag followed by a TEV protease cleavage site. RBBP4 (residues 1–425) and HDAC1 (residues 1–482) were cloned without affinity tags into the same vectors. Protein constructs were co-transfected into HEK293F suspension-grown cells (Invitrogen) with polyethylenimine (PEI; Sigma) and the cells were harvested after 48 hr as previously described (*Nettleship et al., 2014*; *Portolano et al., 2014*; *Watson et al., 2012*). Cells were lysed in 50 mM Tris-HCl (pH 7.5), 100 mM potassium acetate, 10% v/v glycerol, 0.3% v/v Triton X-100 and Roche Complete Protease Inhibitor (buffer A). The clarified lysate was applied to FLAG resin (Sigma) for 2 hr at 4°C. The resin was washed three times with 50 mM Tris-HCl (pH 7.5), 100 mM potassium acetate and 5% v/v glycerol (buffer B), three times with buffer B containing 200 mM potassium acetate and a further three times with buffer B. The protein was treated with RNase A in buffer B for 1 hr at 4°C before being eluted with TEV overnight into buffer B containing 50 mM potassium acetate. The protein complexes were purified further by gel filtration using a Superdex S200 column into 25 mM Tris-HCl (pH 7.5), 50 mM potassium acetate and 0.5 mM TCEP (GE Healthcare, UK). The purified complex was concentrated to 10 mg/ml.

### Gel densitometry

The intensity of protein bands was determined with ImageJ and the ratio of HDAC1:RBBP4 was plotted using GraphPad Prism.

### Size exclusion chromatography with Multi-Angle Light Scattering (MALS)

Purified MTA1:RBBP4, HDAC1:MTA1 and MTA1:HDAC1:RBBP4 complexes that had been gel filtrated were concentrated to 1 mg/ml and reapplied to the Superdex S200 column (GE Healthcare, UK). The mass of each protein complex was calculated immediately on elution with an Optilab T-rEX differential Refractive Index detector coupled to a DAWN HELEOS MALS detector (Wyatt Technology).

### Protein crystallisation

Diffracting crystals (25 μm) of MTA1-R1:RBBP4 were grown at 10°C by sitting-drop vapour diffusion against 0.21 M ammonium citrate and 19% PEG 3350. Crystals were frozen in liquid nitrogen using 30% glycerol as a cryoprotectant, initial screening was performed at the Swiss Light Source PX1 (Switzerland), and data were collected at the Diamond microfocus beamline I24 (UK). Data from two crystals were processed with iMOSFLM (*Leslie, 2005*) and merged using AIMLESS (*Evans and Murshudov, 2013*) to produce a complete dataset. A unique phase solution was found with RBBP4 (pdb code: 4PC0) (*Alqarni et al., 2014*) as a search model using PHASER (*McCoy et al., 2007*) and MTA1 was manually built using multiple rounds of refinement using REFMAC (*Murshudov et al., 2011*) and COOT (*Emsley et al., 2010*). The final model contains four copies of residues 9-411 of RBBP4 (chains A, C, E, G) and residues 468-546 of MTA1 (chains B, D, F, H).

## Small Angle X-ray Scattering (SAXS)

For the SAXS experiments HDAC1:MTA1-A and MTA1-B:HDAC1:RBBP4 were concentrated (1 to 5 mg/ml) in buffer E immediately before analysis. Data were collected on the samples using a Pilatus2M detector (Dectris, CH) at a sample-detector distance of 4014 mm at 12.4 keV at the Diamond B21 beamline (UK). The scattering data was collected to provide a q range of 0.0038-0.42 $Å^{-1}$, where q is the magnitude of the scattering vector, using a flux of c.$10^{11}$ photons per second. The samples and matching buffer solutions were exposed to X-rays 30 times in 5s bursts at 25°C. Examination of the data suggested there was no evidence of radiation damage. The lowest concentration datasets (1 mg/ml) were used for analysis using the ATSAS suite of software (*Petoukhov et al., 2012*) and ScÅtter for basic analysis of the SAXS datasets (BioIsis.net). The radius-of-gyration (Rg) was calculated from Guinier plots and these values were in good agreement with the Rg calculated from the second moment of the P(r) function. The maximum dimension of the scattering molecular species (Dmax) was estimated from the P(r) curve and again these values were favourable comparable to those calculated using the program GNOM (*Svergun, 1992*). Theoretical scattering profiles were calculated from the crystal structure of HDAC1:MTA1 (*Millard et al., 2013*) and combined crystal structures of HDAC1:MTA1 and MTA1-R1:RBBP4 and compared to the experimental data using FoXS (*Schneidman-Duhovny et al., 2010*). Thirteen a*b initio* dummy atom models were generated using the programme DAMMIF (*Franke and Svergun, 2009*) and were superposed, merged and filtered using the programme DAMAVER (*Volkov and Svergun, 2003*).

## Chemical crosslinking

Gel filtrated MTA1-B:HDAC1:RBBP4 complex was buffer exchanged into 50 mM HEPES (pH 7.5), 50 mM potassium acetate, diluted to 0.5 mg/ml, and the isotopically labelled crosslinker CBDPSS-H8/D8 (CyanurBiotinDimercaptoPropionylSulfoSuccinimide, Creative Molecules Inc.) (*Petrotchenko et al., 2011*) was added to a final concentration of 0.7 mM. After incubation at 30°C for 30 min the reaction was stopped by addition of 0.5 M $NH_4HCO_3$ to a final concentration of 40 mM. The protein was analysed by SDS-PAGE and protein bands corresponding to the cross-linked-dimeric complex were excised from the gel. The sample was reduced with 10 mM DTT, alkylated with 100 mM iodoacetamide and digested with trypsin.

LC-MS/MS was carried out as described previously (*Alam et al., 2015*). Raw data files were converted to. mzXML format using the ProteoWizard msconvert toolkit (*Chambers et al., 2012*) and cross-linked peptides were identified using the xQuest/xProphet (*Leitner et al., 2014*) pipeline with appropriate parameters defined for the CBDPSS isotopically labelled crosslinker. For crosslinks to be considered valid the xQuest Id-Score was required to exceed 14.5, and on inspection of the MS/MS spectra a minimum of either four fragment ions per peptide or three consecutive fragment ions per peptide were required to match.

## EM sample preparation

Gel filtrated MTA1-B (162-546):HDAC1:RBBP4 was prepared for EM analysis by using GraFIX (*Kastner et al., 2007*). A 5-25% sucrose gradient with 0-1% glutaraldehyde in 20 mM HEPES/KOH (pH 7.5), 40 mM NaCl was prepared using a Gradient Master IP (BIOCOMP) with a SW60 rotor. The protein complex was added directly on top of the gradient and centrifuged for 15 hr at 166400g in a swing-out rotor. The gradient was manually fractionated into 175 µl fractions and fractions containing monodisperse complex were pooled. The sucrose was removed and cross linking stopped by buffer exchange in a centrifugal filter concentrator (Millipore, Amicon, Ultra-0.5, 10 kDa cut-off) using 20 mM Tris/HCl (pH 7.5), 40 mM NaCl.

## EM data collection

Negative stain grids were prepared by glow-discharging carbon coated copper 300 mesh grids (Agar Scientific) at 10 mA for 30 s. The protein complex was diluted to 0.1 mg/ml and 5 µl was applied to the grid. The protein was allowed to absorb for 1 min and blotted to remove excess liquid. The grid was then stained with 2% w/v uranyl acetate. Grids were imaged using a Jeol 2010F TEM operating at 200 kV fitted with a Gatan UltraScan™ 4000 CCD camera with a 15 µm pixel size. 308 micrographs were collected at various defocus values between 0.5 µm and 5 µm, each micrograph contained 20 – 90 particles.

## EM image processing

17,841 particles were semi-automatically picked using the EMAN2 e2boxer.py swarm function and extracted using a 224 × 224 pixel box size. Reference-free 2D class averages were generated, with c.100 – 200 particles per class average using e2refine2d.py (*Tang et al., 2007*). A 60 Å filtered model of the complex was generated based on the crystal structures of HDAC1:MTA1 and MTA1: RBBP4 fitted to the SAXS and crosslinking data, refined against the 17,841 particle dataset, and a model was generated with C1 symmetry using e2refine_easy.py. Resolution of refined model was 19 Å using a FSC criterion cutoff of 0.143.

## Peptide binding assays

Gel filtrated Flag-RBBP4 and Flag-MTA1-R1-:RBBP4 proteins were diluted to 200 nM in 5 ml TTBS (10 mM Tris-HCl (pH 7.4), 0.05% Tween 20 and 150 mM NaCl) and applied to separate MODified histone peptide arrays (Active Motif) containing 384 unique histone modification combinations. The arrays were washed and protein detected using an anti-Flag antibody according to the manufacture's protocol. Data from each array was analysed using the commercially available software (Active Motif).

## Acknowledgements

We would like to thank Xiaowen Yang (PROTEX) for cloning all the expression constructs used in the study. We are grateful to beamline scientists Takashi Tomizaki (PX1, SLS), James Doutch (B21, Diamond) and Neil Paterson (I24, Diamond) for their help in screening crystals, collecting SAXS and X-ray data respectively. We would also like to thank Khushwant Sidhu for assistance with installing and running ProteoWizard msconvert and xQuest/xProphet software. We thank Ian Hands-Portman for technical support with electron microscopy and the Wellcome Trust for generous support (055663/Z/98/Z) to the Imaging Suite at The University of Warwick. CJS and KM thank BBSRC (BB/K003461/1) for support. JWRS is supported by a Senior Investigator Award (WT100237) from the Wellcome Trust and a Biotechnology and Biological Sciences Research Council Project Grant (BB/J009598/1). JWRS is a Royal Society Wolfson Research Merit Award Holder.

## Additional information

### Funding

| Funder | Grant reference number | Author |
|---|---|---|
| Wellcome Trust | WT100237 | John WR Schwabe |
| Biotechnology and Biological Sciences Research Council | BB/J009598/1 | John WR Schwabe |
| Royal Society | Wolfson Research Merit Award | John WR Schwabe |
| Wellcome Trust | 055663/Z/98/Z | Corinne J Smith |
| Biotechnology and Biological Sciences Research Council | BB/K003461/1 | Kyle Morris Corinne J Smith |

The funders had no role in study design, data collection and interpretation, or the decision to submit the work for publication.

### Author contributions

CJM, NV, LF, Conception and design, Acquisition of data, Analysis and interpretation of data, Drafting or revising the article; AS, KM, Acquisition of data, Analysis and interpretation of data, Drafting or revising the article; PJW, CJS, Analysis and interpretation of data, Drafting or revising the article; ARB, Acquisition of data, Drafting or revising the article; JWRS, Conception and design, Analysis and interpretation of data, Drafting or revising the article

Author ORCIDs

Christopher J Millard, http://orcid.org/0000-0002-1012-0829
Almutasem Saleh, http://orcid.org/0000-0002-0156-4508
Andrew R Bottrill, http://orcid.org/0000-0002-5182-3643
John WR Schwabe, http://orcid.org/0000-0003-2865-4383

## Additional files

### Major datasets

The following datasets were generated:

| Author(s) | Year | Dataset title | Dataset URL | Database, license, and accessibility information |
|---|---|---|---|---|
| Millard CJ, Varma N, Fairall L, Schwabe JWR | 2016 | Structure of the Human RBBP4: MTA1(464-546) complex | http://www.rcsb.org/pdb/search/structid-Search.do?structureId=5FXY | Publicly available at RCSB Protein Data Bank (accession no. 5fxy) |
| Millard CJ, Saleh A, Morris K, Fairall L, Smith CJ, Schwabe JWR | 2016 | Structure of the core NuRD complex (MTA1:HDAC1:RBBP4) | http://www.ebi.ac.uk/pdbe/entry/emdb/EMD-3399 | Publicly available at EMData Bank (accession no. EMD-3399) |

The following previously published dataset was used:

| Author(s) | Year | Dataset title | Dataset URL | Database, license, and accessibility information |
|---|---|---|---|---|
| Millard CJ, Watson PJ, Celardo I, Gordiyenko Y, Cowley SM, Robinson CV, Fairall L, Schwabe JWR | 2013 | The structure of HDAC1 in complex with the dimeric ELM2-SANT domain of MTA1 from the NuRD complex | http://www.rcsb.org/pdb/explore/explore.do?structureId=4BKX | Publicly available at RCSB Protein Data Bank (accession no. 4bkx) |

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
