## [Decision Letter]

Thank you for submitting your work entitled "The structure and stoichiometry of RBBP4 in the NuRD complex suggests a mechanism of recruitment of HDAC1/2 to chromatin" for consideration by *eLife*. Your article has been favorably evaluated by Kevin Struhl (Senior Editor) and three reviewers, one of whom, Jerry Workman, is a member of our Board of Reviewing Editors, and another is Paul Wade.

The reviewers have discussed the reviews with one another and the Reviewing Editor has drafted this decision to help you prepare a revised submission.

Summary:

The manuscript reports on the stoichiometry and structure of the MTA1:RPBB4 interaction and its implications for the organization and chromatin interaction of the NuRD complex. The authors convincingly confirm previous reports of a possible second RPBB4 interaction site in MTA1 and of a 1:2 stoichiometry for the MTA1:RBBP4 complex (both of these previous reports are appropriately referenced). An X-ray structure of RBBP4 interacting with a ~80 amino acid segment of MTA1 containing the novel/previously suspected binding site provides an improved understanding of a possibly conserved MTA1:RBBP4 interaction mode. The experiments to test interaction of RBBP4 with various MAT1 segments are convincing and the implications from analysis of the X-ray structure showing the "new" interaction with MAT1 are interesting.

Essential revisions:

Overall the reviewers support the study and its eventual publication. However, there were two major issues that were deemed important to address. These major issues are numbered below and discussed in some detail. The first issue will require some additional analysis of your EM data and perhaps some new EM data. The second can be addressed by editing the Results and Discussion.

1) Based on the observations above, their published X-ray structure of the MTA1 dimer, and sequence analysis of MTA1, the authors propose that the MTA1:HDAC1:RBBP4 proteins could form the structural core of the NuRD complex and set out to determine the structural organization of that core. MALS was used to analyze the stoichiometry and SAXS to investigate the core's overall structure. These results were complemented by cross-linking/mass spectrometry and analysis of single particle electron microscopy (EM) images of the core. The MALS results to test complex stoichiometry, and the SAXS results to evaluate overall complex structure are sound. However, the cross-linking/MS and EM results used to derive a structure for the MTA1:HDAC1:RBBP4 are not as convincing. The SAXS envelope is, as expected very low resolution and cannot determine more than the overall position of RBBP4. The authors relied on the MS analysis to orient RBBP4 in the complex and then compared views of their cross-linking based model with class-averages obtained from EM images. About two third of crosslinks are consistent with the proposed model and the authors highlight a number of EM class averages that also appear consistent with the model. This seems sufficient to validate the proposed model, but actually calculating a low-resolution structure from the EM images would have provide more definite evidence. Since the EM class averages seem to show various views of the complex, it is conceivable that those averages contain enough information to calculate a 3D map. There are also other simple approaches that could have been used to obtain an actual structure from EM images of stained particles.

A second issue worth considering is the possibility that, even if the interpretation of the current data is entirely correct, in a larger complex (e.g., a full complement of RBBP4s, other NuRD components), additional interactions between subunits could result in an organization of MAT1:HDAC1:RBBP4 entirely different from the one depicted in the final model. If NuRD can be purified to reasonable homogeneity, it would be worth collecting some EM images in stain and rule out the possibility that the structure of the complete NuRD, and therefore its integration with chromatin, are different from what is proposed here. If it is not possible to record images of NuRD and confirm that part of NuRD indeed looks like the averages in Figure 4, maybe a reasonable compromise would be to indicate in the Discussion the possible limitations of the model for core organization and chromatin interaction shown in Figure 6.

2) While I applaud the dataset, the text, as written, fails to discuss critical features revealed in the data.

First, could the authors please discuss whether the RBBP4/7 structure in the complex with MTA1/HDAC1 differs from the published structure? This is an important detail.

Second, the binding pattern depicted in the peptide array indicates a degree of specificity that would make the manuscript more interesting to a general audience. If I am not mistaken, the array indicates complex interaction with H3 amino acids 16-35 that appears to require modification of lysine 27. I had to look up the positions of peptides on the array to discover that (please correct me if I am wrong). Given the biology around K27 Ac/Me choice and NuRD complex as described by Brian Hendrich, this finding is quite interesting. I am perplexed as to why this is not discussed in the current manuscript.

Finally, it is not clear how the binding pattern would differ if the authors included both R1 and R2 in the construct. Has this experiment been done?

---

## [Author Response]

Essential revisions:

*Overall the reviewers support the study and its eventual publication. However, there were two major issues that were deemed important to address. These major issues are numbered below and discussed in some detail. The first issue will require some additional analysis of your EM data and perhaps some new EM data. The second can be addressed by editing the Results and Discussion. 1) Based on the observations above, their published X-ray structure of the MTA1 dimer, and sequence analysis of MTA1, the authors propose that the MTA1:HDAC1:RBBP4 proteins could form the structural core of the NuRD complex and set out to determine the structural organization of that core. MALS was used to analyze the stoichiometry and SAXS to investigate the core's overall structure. These results were complemented by cross-linking/mass spectrometry and analysis of single particle electron microscopy (EM) images of the core. The MALS results to test complex stoichiometry, and the SAXS results to evaluate overall complex structure are sound. However, the cross-linking/MS and EM results used to derive a structure for the MTA1:HDAC1:RBBP4 are not as convincing. The SAXS envelope is, as expected very low resolution and cannot determine more than the overall position of RBBP4. The authors relied on the MS analysis to orient RBBP4 in the complex and then compared views of their cross-linking based model with class-averages obtained from EM images. About two third of crosslinks are consistent with the proposed model and the authors highlight a number of EM class averages that also appear consistent with the model. This seems sufficient to validate the proposed model, but actually calculating a low-resolution structure from the EM images would have provide more definite evidence. Since the EM class averages seem to show various views of the complex, it is conceivable that those averages contain enough information to calculate a 3D map. There are also other simple approaches that could have been used to obtain an actual structure from EM images of stained particles.*

We thank the reviewers for this suggestion. We have now used 17,841 automatically-picked particles to calculate class averages and a low-resolution (c.19Å) structure of the complex. This structure fits well with the SAXS and cross-linking data and also provides additional information allowing us to refine the relative orientations of the RBBP4:MTA1 and HDAC1:MTA1 structures. This structure suggests a somewhat tighter arrangement of HDAC1 and RBBP4 than the model based on the SAXS and crosslinking data. This structure is presented in the text and in Figure 4. Details of the methodology are also included in the Methods and supplementary material.

*A second issue worth considering is the possibility that, even if the interpretation of the current data is entirely correct, in a larger complex (e.g., a full complement of RBBP4s, other NuRD components), additional interactions between subunits could result in an organization of MAT1:HDAC1:RBBP4 entirely different from the one depicted in the final model. If NuRD can be purified to reasonable homogeneity, it would be worth collecting some EM images in stain and rule out the possibility that the structure of the complete NuRD, and therefore its integration with chromatin, are different from what is proposed here. If it is not possible to record images of NuRD and confirm that part of NuRD indeed looks like the averages in Figure 4, maybe a reasonable compromise would be to indicate in the Discussion the possible limitations of the model for core organization and chromatin interaction shown in Figure 6.* We agree that it is possible that the relative arrangement of HDAC1, MTA1 and RBBP4 may be somewhat different in the holo-NuRD complex. We have mentioned this potential limitation in the discussion. However, the relatively close interaction between HDAC1 and RBBP4, seen in the EM structure, suggests to us that this interface is likely also present in the holo-complex.

We have been trying to prepare the holo-NuRD complex, but have not been successful as yet. Unfortunately, we are therefore unable to collect images of the fully assembled NuRD complex.

*2) While I applaud the dataset, the text, as written, fails to discuss critical features revealed in the data.*

*First, could the authors please discuss whether the RBBP4/7 structure in the complex with MTA1/HDAC1 differs from the published structure? This is an important detail.*

We have now clearly described the differences between the structures of RBBP4 and our MTA1-R1:RBBP4 complex in the Results section. In brief, there are relatively few changes in the structure of RBBP4 except for an extended loop that wraps around MTA1-R1 in the complex.

*Second, the binding pattern depicted in the peptide array indicates a degree of specificity that would make the manuscript more interesting to a general audience. If I am not mistaken, the array indicates complex interaction with H3 amino acids 16-35 that appears to require modification of lysine 27. I had to look up the positions of peptides on the array to discover that (please correct me if I am wrong). Given the biology around K27 Ac/Me choice and NuRD complex as described by Brian Hendrich, this finding is quite interesting. I am perplexed as to why this is not discussed in the current manuscript.*

Finally, it is not clear how the binding pattern would differ if the authors included both R1 and R2 in the construct. Has this experiment been done?

We are grateful for this suggestion to discuss the results of binding to the modified histone peptide array in more detail. We have added a paragraph in the Discussion highlighting the interaction between RBBP4 and histone H3 and the apparent requirement for modification at H3K27 – particularly acetylation, but also methylation. We have also added a "key" to the array in Figure 6—figure supplement 2, Table 1.